# Ethnicity-Specific Molecular Alterations in MAPK and JAK/STAT Pathways in Early-Onset Colorectal Cancer

**DOI:** 10.3390/cancers17071093

**Published:** 2025-03-25

**Authors:** Cecilia Monge, Brigette Waldrup, Francisco G. Carranza, Enrique Velazquez-Villarreal

**Affiliations:** 1Center for Cancer Research, National Cancer Institute, Bethesda, MD 20892, USA; 2City of Hope, Beckman Research Institute, Department of Integrative Translational Sciences, Duarte, CA 91010, USA; 3City of Hope Comprehensive Cancer Center, Duarte, CA 91010, USA

**Keywords:** early-onset colorectal cancer, cancer disparities, genetic mutations, precision medicine, MAPK pathway, JAK/STAT pathway

## Abstract

Early-onset colorectal cancer (EOCRC), diagnosed before age 50, is becoming more common, particularly among Hispanic/Latino (H/L) individuals. However, the biological reasons behind this increase are not well understood. This study examines genetic changes in two key pathways—MAPK and JAK/STAT, which influence cancer growth and treatment response—to determine whether these changes differ between H/L and Non-Hispanic White (NHW) patients. Using data from over 3400 colorectal cancer patients, the researchers found that H/L EOCRC patients had higher rates of specific MAPK pathway alterations compared to NHW patients, while no major differences were observed in the JAK/STAT pathway. Additionally, survival outcomes were influenced by alterations in these pathways, particularly in NHW patients. These findings highlight the need for further research into ethnic differences in tumor biology and suggest that personalized treatment strategies could improve outcomes for H/L EOCRC patients.

## 1. Introduction

Ranked as the third most common cancer globally, colorectal cancer (CRC) is also a leading cause of cancer-related mortality, placing second in fatalities [1,2]. Although the overall incidence of CRC has remained stable or decreased in higher-income nations, the global surge in early-onset colorectal cancer (EOCRC), diagnosed before age 50, has become a growing concern [1,3,4,5]. This trend is particularly pronounced in the Hispanic/Latino (H/L) population, which has experienced the most significant increase in EOCRC incidence among all racial and ethnic groups in the United States [6,7]. Furthermore, the rise in EOCRC-related mortality is most pronounced among Hispanic/Latino (H/L) individuals, surpassing that of non-Hispanic White (NHW) populations and underscoring a significant health disparity [8,9,10]. These disparities underscore the urgent need for a deeper understanding of the molecular mechanisms contributing to CRC development and progression in H/L patients.

Screening guidelines that recommend routine CRC surveillance beginning at age 50 may contribute to late-stage diagnoses in EOCRC patients [11]. Beyond delayed detection, emerging research suggests that EOCRC has distinct molecular characteristics compared to late-onset colorectal cancer (LOCRC), including a higher prevalence of microsatellite instability (MSI), an increased tumor mutation burden, and elevated expression of immune checkpoint regulators such as PD-L1 [12,13,14,15]. Additionally, epigenetic alterations, such as LINE-1 hypomethylation, have been proposed as biomarkers that distinguish EOCRC from LOCRC [16]. These findings suggest that EOCRC may develop through distinct oncogenic pathways, justifying further investigation into ethnicity-specific genomic alterations.

In CRC pathogenesis, the MAPK and JAK/STAT pathways play essential roles by regulating cell proliferation, differentiation, survival, and immune responses through key signaling cascades. The MAPK pathway comprises four primary signaling families—ERK, BMK-1, JNK, and p38—all of which regulate key oncogenic processes [16]. In CRC, aberrant MAPK signaling is frequently driven by oncogenic mutations in RAS and BRAF, leading to uncontrolled tumor growth, resistance to apoptosis, and metastasis [17,18]. Notably, BRAF mutations, which are common in CRC, have been shown to confer resistance to RAF inhibitors, such as vemurafenib, due to reactivation of the MAPK pathway through EGFR signaling [19,20]. While the role of the MAPK pathway in CRC is well established [21,22,23,24,25,26], its contribution to EOCRC disparities in H/L patients remains poorly understood.

The JAK/STAT pathway is another essential signaling cascade that mediates cellular responses to inflammation, an established driver of CRC progression [27,28,29]. Persistent activation of JAK/STAT signaling has been implicated in tumor progression, therapy resistance, and poor clinical outcomes in CRC patients [29,30,31]. Elevated JAK/STAT activity has been associated with increased cytokine production, enhanced immune evasion, and epithelial-to-mesenchymal transition (EMT) in CRC, ultimately promoting tumor aggressiveness [29]. Additionally, STAT3 activation in CRC [32] has been linked to stromal invasion and reduced cancer-specific survival [33], particularly in tumors with high inflammatory signaling [34,35,36]. Despite its significance, the role of JAK/STAT pathway alterations in EOCRC, particularly among H/L patients, remains unexplored.

Given the increasing burden of EOCRC in the H/L population and the limited knowledge of ethnicity-specific oncogenic mechanisms, this study aims to characterize molecular alterations in the MAPK and JAK/STAT pathways (Figure 1) in EOCRC. By comparing EOCRC in H/L and NHW patients, we seek to identify key pathway-specific differences that may contribute to CRC disparities. Moreover, we analyze how these molecular alterations affect patient survival, providing valuable perspectives on possible disease indicators and treatment approaches. Gaining deeper insight into the genetic and cellular framework of EOCRC in underrepresented populations is vital for refining personalized treatment strategies and improving prognoses for H/L individuals with CRC.

## 2. Materials and Methods

Clinical and genomic data from all CRC datasets available in the cBioPortal database, at the time of this study, were analyzed. The analysis included datasets on colorectal, colon, and rectal adenocarcinomas, along with data from the GENIE BPC CRC v2.0-open-access dataset. To ensure a focus on primary tumor cases, two datasets that specifically examined metastatic CRC were excluded. Patients were selected based on predefined inclusion criteria, which required identification as Hispanic or Latino, Spanish, NOS (not otherwise specified); Hispanic, NOS; Latino, NOS; or individuals with a Mexican or Spanish surname. Additional filtering criteria included limiting cases to primary tumors, selecting colorectal, colon, and rectal adenocarcinomas, confirming adenocarcinoma, NOS histology, and ensuring only one sample per patient. Out of all the databases, three met the specified selection criteria, yielding a cohort of 302 H/L patients (138 EOCRC and 164 LOCRC). Similarly, 3110 NHW patients (897 EOCRC and 2213 LOCRC) were identified using the same inclusion criteria (Table 1 and Table 2). Age at diagnosis was obtained from GENIE database clinical records. This study represents one of the largest investigations into MAPK and JAK/STAT pathway alterations in an underserved population, providing essential insights into the molecular disparities between EOCRC and LOCRC patients.

Previously defined guidelines were used to determine pathway alterations [18] and perform the analyses [37,38]. This study primarily examines genetic alterations rather than changes in gene expression. Patients were categorized into EOCRC (<50 years of age) and LOCRC (≥50 years) groups, with further stratification based on ethnicity (H/L vs. NHW) and pathway status (presence or absence of MAPK and JAK/STAT pathway alterations). Table 3 presents the distribution of EOCRC and LOCRC H/L CRC patients, highlighting the prevalence of MAPK and JAK/STAT pathway alterations in each group. Table 4 extends this analysis by comparing EOCRC H/L and EOCRC NHW patients, facilitating a comparative assessment of pathway-specific differences between ethnic groups. Through these stratifications, this study offers a comprehensive molecular characterization of MAPK and JAK/STAT pathway alterations, providing valuable insights into potential ethnicity-specific disparities that could inform precision medicine strategies for CRC.

To compare somatic mutation frequencies between groups, Chi-square tests were conducted, assessing the independence of categorical variables and analyzing associations among age, ethnicity, and pathway alterations. Additionally, tumor samples were further stratified based on tumor location (colon vs. rectal adenocarcinoma), allowing for a more refined examination of the interaction between tumor site, ethnicity, and pathway alterations. This degree of classification enabled a nuanced investigation into patient distinctiveness and possible influences on treatment strategies.

To assess the prognostic relevance of MAPK and JAK/STAT pathway alterations, survival probabilities were analyzed using the Kaplan–Meier approach. across different patient subgroups. Survival curves were constructed to illustrate differences in survival probability over a period, with patients classified based on whether pathway alterations were present or not. To assess statistically significant differences between survival curves, the log-rank test was conducted, while median survival times and 95% confidence intervals were estimated. By integrating large-scale genomic data, survival analysis, and stratified comparisons, this study provides a detailed assessment of pathway-specific alterations in EOCRC and LOCRC patients, particularly within the H/L population, offering new insights into ethnicity-specific molecular mechanisms in CRC.

Our analysis focused exclusively on somatic mutations, as opposed to germline polymorphisms, to ensure that the identified variants were relevant to tumorigenesis. All mutations were extracted from publicly available cancer-specific datasets, which primarily catalog alterations associated with colorectal cancer progression. Where applicable, we annotated variant classifications, including Frame Shift Deletion, Frame Shift Insertion, In-Frame Deletion, In-Frame Insertion, Missense Mutation, Nonsense Mutation, Splice Region, Splice Site, and Translation Start Site, to provide greater clarity on their potential impact on tumor biology. These measures enhance the interpretability and biological relevance of our results while minimizing ambiguity in distinguishing tumor-driving mutations from inherited polymorphisms.

## 3. Results

We established our H/L cohort, consisting of 3412 samples, using data from three cBioPortal projects that reported ethnicity, as described in Table 1. In contrast, the NHW cohort had 16.2% EOCRC patients, with the majority (83.8%) reported at 50 years of age or greater. The gender distribution within the H/L cohort included 58.3% male and 41.7% female patients, whereas the NHW cohort consisted of 55% male and 45% female participants. Upon identification, 32.5% of participants in the H/L group presented with stage 0, I, II, or III disease, while stage IV cancer was observed in 43.7% of cases. In comparison, the NHW cohort had 31% of patients diagnosed at stages 0, I, II, or III, and 36.4% at stage IV. Notably, for 23.8% of H/L patients and 32.6% among NHW participants, stage at diagnosis was documented as NA, indicating missing or unreported staging information. Ethnicity classification within the H/L cohort revealed that the majority of patients (89.4%) identified as Spanish NOS, Hispanic NOS, or Latino NOS, while 9.3% were classified as Mexican (including Chicano), and 1.3% identified as either Other Spanish/Hispanic or Spanish surname only. Conversely, all patients in the NHW cohort (100%) were classified as NHW, ensuring a clear distinction between the two groups for comparative analyses.

A comparison of clinical features between EOCRC and LOCRC H/L patients, as well as between EOCRC H/L and EOCRC NHW patients, identified significant distinctions (Table 2). The median age at diagnosis for EOCRC H/L patients was 41 years (IQR: 36–46), significantly younger than the median of 61 years (IQR: 55–69) observed in LOCRC H/L patients (*p* < 0.05). Similarly, EOCRC H/L patients were diagnosed at a significantly younger age compared to EOCRC NHW patients (median: 42 years, IQR: 38–47) (*p* < 0.05). Analysis of mutation burden revealed a median mutation count of 7 in EOCRC H/L patients, compared to a slightly higher median count of 8 in LOCRC H/L patients; however, this difference was not statistically significant (*p* > 0.05). Similarly, EOCRC NHW patients had a median mutation count of 7, comparable to EOCRC H/L patients, with no statistically significant difference between these groups (*p* > 0.05).

Regarding pathway-specific alterations, several genes within the MAPK pathway were found to be more frequently mutated in EOCRC H/L patients than in their LOCRC counterparts. Notably, NF1 (11.6% vs. 3.7%, *p* = 0.01), ACVR1 (2.9% vs. 0%, *p* = 0.04), and MAP2K1 (3.6% vs. 0%, *p* = 0.01) exhibited significantly higher mutation frequencies in EOCRC. In contrast, BRAF mutations were significantly more common in LOCRC H/L patients compared to EOCRC patients (18.3% vs. 5.1%, *p* = 9.1 × 10^−4^). When comparing EOCRC in H/L versus NHW patients, key MAPK pathway genes such as AKT1 (5.1% vs. 1.8%, *p* = 0.03), MAPK3 (3.6% vs. 0.7%, *p* = 6.83 × 10^−3^), NF1 (11.6% vs. 6.1%, *p* = 0.02), and PDGFRB (5.8% vs. 2.1%, *p* = 0.02) were significantly enriched in EOCRC H/L patients. These findings underscore potential ethnicity-specific genomic differences that may contribute to variations in CRC pathogenesis.

The observed disparities in MAPK pathway alterations between EOCRC and LOCRC, as well as between H/L and NHW EOCRC patients, highlight the need for further investigation into the biological and clinical implications of these alterations. Understanding these differences may help inform targeted therapeutic strategies aimed at addressing disparities in CRC outcomes across diverse ethnic populations.

As part of our examination of genetic variations among H/L individuals with EOCRC and LOCRC, no significant differences were observed in the frequency of MAPK and JAK/STAT pathway alterations (Table 3). JAK/STAT pathway alterations were present in 11.6% of both EOCRC and LOCRC patients, with an identical absence rate of 88.4% in both groups. Similarly, MAPK pathway alterations were highly prevalent in both groups, occurring in 97.1% of early-onset patients and 98.2% of late-onset patients, though this difference was not statistically significant (*p* = 0.7). The absence of MAPK pathway alterations was slightly more common in EORCR patients (2.9%) compared to LOCRC patients (1.8%). These findings suggest that MAPK and JAK/STAT pathway alterations are consistently frequent across both age groups, indicating that their role in CRC development may not be significantly influenced by age at onset in H/L patients. Further studies are justified to explore potential interactions between these pathways and other molecular drivers contributing to CRC progression in this population.

In our analysis of genetic alterations in EOCRC among H/L and NHW individuals, no statistically significant differences were observed in the frequency of MAPK and JAK/STAT pathway alterations (Table 4). JAK/STAT alterations were detected in 11.6% of EOCRC H/L patients compared to 8.7% of EOCRC NHW patients (*p* = 0.34). The absence of JAK/STAT alterations was slightly more common in NHW individuals (91.3%) than in H/L individuals (88.4%), though this difference was not statistically significant (*p* = 0.34). Similarly, MAPK pathway alterations were highly prevalent in both EOCRC groups, occurring in 99.3% of H/L patients and 96% of NHW patients (*p* = 0.66). The absence of MAPK pathway alterations was slightly more frequent in NHW individuals (4%) than in H/L individuals (2.9%), but this difference also lacked statistical significance (*p* = 0.34). These findings suggest that MAPK and JAK/STAT pathway alterations may be marginally more frequent in EOCRC H/L individuals than in their NHW counterparts, but the observed differences are not statistically significant. Further analyses incorporating larger cohorts and functional studies are needed to better understand the clinical relevance of these pathway alterations and their potential implications for precision medicine and targeted therapies in CRC.

The survival analysis using the Kaplan–Meier method conducted for EOCRC H/L CRC patients showed no statistically significant difference in overall survival between those with and without MAPK pathway alterations (Figure 2). Although the survival curves for both groups exhibited slight divergence over time, the *p*-value (*p* = 0.9) indicated that this variation was not statistically meaningful. The confidence intervals around each curve highlight the variability in survival estimates at different time points, emphasizing the uncertainty of these results. These results suggest that MAPK pathway alterations may not play a major role in influencing overall survival outcomes in EOCRC H/L patients. Further studies with larger sample sizes and additional molecular stratifications are needed to better understand the potential impact of MAPK pathway alterations in this population.

The Kaplan–Meier survival analysis for EOCRC H/L CRC patients with and without JAK/STAT pathway alterations (Figure 2) revealed a trend toward worse survival outcomes in patients with JAK/STAT alterations. Although the difference in overall survival did not reach statistical significance, the *p*-value (*p* = 0.073) suggests a borderline association. Patients with JAK/STAT pathway alterations exhibited an early decline in survival probability, with consistently lower survival rates throughout the follow-up period compared to those without alterations. In contrast, patients without JAK/STAT alterations demonstrated a more gradual decline in survival probability, maintaining higher overall survival rates over time. These findings indicate a potential prognostic impact of JAK/STAT pathway alterations in EOCRC H/L patients, justifying further investigation. Larger cohorts and additional molecular characterizations are necessary to clarify the role of JAK/STAT alterations in survival outcomes within this population.

Similar to the results observed for JAK/STAT pathway in the H/L cohort, the overall survival analysis for the NHW cohort (Appendix A) suggests that JAK/STAT pathway alterations significantly impact survival outcomes in EOCRC within this ethnic group. The highly significant *p*-value (<0.0001) indicates a strong association between JAK/STAT pathway alterations and poorer survival, highlighting its potential prognostic relevance. In contrast, MAPK pathway alterations in NHW patients did not show a statistically significant difference in overall survival, with a *p*-value of 0.49. These findings contrast with those observed in the H/L cohort, where neither MAPK nor JAK/STAT alterations reached statistical significance for survival differences. This suggests that JAK/STAT pathway alterations may play a more prominent role in EOCRC survival among NHW patients, emphasizing the need for further investigation into ethnicity-specific molecular drivers of CRC prognosis.

To identify potential age-related variations, the alteration rates of genes related to the MAPK and JAK/STAT pathways were examined among EOCRC and LOCRC H/L patients (Appendix A). The findings indicated that specific MAPK pathway genes, including NF1, ACVR1, and MAP2K1, were significantly more prevalent in EOCRC patients compared to LOCRC patients. NF1 alterations were observed in 11.6% of EOCRC patients, compared to only 3.7% of LOCRC patients (*p* = 0.01). Similarly, ACVR1 and MAP2K1 alterations were exclusive to EOCRC patients (2.9% and 3.6%, respectively) and were absent in LOCRC patients (*p* = 0.04 and *p* = 0.01, respectively). Conversely, BRAF alterations were significantly more frequent in LOCRC patients (18.3%) compared to EOCRC patients (5.1%, *p* = 9.1 × 10^−4^), indicating a potential age-associated pattern in MAPK pathway gene alterations.

Comparing EOCRC H/L patients to their NHW counterparts, several key MAPK pathway genes exhibited significantly higher alteration rates in H/L patients (Appendix A). AKT1 mutations were more frequent in EOCRC H/L patients (5.1%) compared to EOCRC NHW patients (1.8%, *p* = 0.03). Similarly, MAPK3 (3.6% vs. 0.7%, *p* = 6.83 × 10^−3^), NF1 (11.6% vs. 6.1%, *p* = 0.02), and PDGFRB (5.8% vs. 2.1%, *p* = 0.02) showed a significantly higher prevalence in EOCRC H/L patients. In contrast, JAK/STAT pathway alterations did not show significant differences between EOCRC and LOCRC H/L patients or when comparing EOCRC H/L and NHW patients. These findings suggest that while alterations in the MAPK pathway may play a distinct role in EOCRC CRC among H/L patients, JAK/STAT pathway alterations appear to be less associated with age or ethnicity-specific differences. The significantly higher frequency of NF1, ACVR1, MAP2K1, and AKT1 mutations in EOCRC H/L patient data support the call for expanded investigation into their biological significance in tumor development and treatment response. Future studies should explore the implications of these molecular differences in precision medicine strategies tailored for H/L CRC patients across different age groups.

Analysis of MAPK pathway alterations in H/L EOCRC patients revealed a diverse spectrum of mutation types, as summarized in Appendix A. The majority of mutations identified were missense mutations, which were predominant across most genes, including ACVR1 (100%), BRAF (100%), CRKL (100%), FGFR1 (100%), HRAS (100%), KRAS (100%), MAP2K1 (100%), MAP2K2 (100%), MAP3K13 (100%), MAPK1 (100%), MAPK3 (100%), NRAS (100%), NTRK1 (100%), NTRK2 (100%), RAC1 (100%), RAF1 (100%), and RRAS (100%). This suggests that single amino acid substitutions are a major mode of alteration within the MAPK pathway in this population. Other mutation types were observed at lower frequencies. Frame shift deletions were identified in genes such as AKT2 (100%), DAXX (28.6%), EGFR (20%), FGF3 (50%), FGFR4 (33.3%), JUN (40%), NF1 (21.2%), RASA1 (50%), RPS6KA4 (14.3%), TGFBR2 (41.7%), and TP53 (6.4%), indicating potential disruptions in protein structure. Notably, nonsense mutations, which result in premature stop codons, were detected in AKT3 (40%), FGFR2 (22.2%), MAP2K4 (33.3%), PDGFRB (10%), and TP53 (18.4%), suggesting possible loss-of-function effects in these genes. Additionally, splice site mutations were observed in EGFR (20%), FGFR4 (16.7%), MAP2K4 (16.7%), NF1 (9.1%), PDGFRA (7.1%), and TP53 (6.4%), potentially impacting normal splicing and gene expression regulation. These findings provide a detailed characterization of the nature of MAPK pathway mutations in H/L EOCRC patients, emphasizing the predominance of missense mutations alongside structural alterations that may influence tumor progression and treatment response. The variability in mutation types highlights the need for further functional studies to determine their role in colorectal cancer disparities and precision medicine approaches for H/L populations.

Examination of JAK/STAT pathway alterations in H/L EOCRC patients revealed a heterogeneous distribution of mutation types, as detailed in Appendix A. Missense mutations were the most frequently observed alteration type, particularly in JAK2 (100%), JAK3 (75%), SOCS1 (100%), and STAT3 (50%), suggesting that single amino acid substitutions may play a significant role in modulating pathway activity in H/L EOCRC. Other structural mutations were also present, with frame shift deletions and insertions detected in JAK1 (12.5% deletion, 6.3% insertion), STAT5A (50% deletion, 50% insertion), and STAT5B (50% deletion, 33.3% insertion). These mutations have the potential to disrupt protein function by altering the reading frame, which may lead to truncated or dysfunctional proteins within the JAK/STAT signaling cascade. Additionally, nonsense mutations, which introduce premature stop codons and may lead to loss-of-function effects, were identified in JAK1 (12.5%) and JAK3 (25%). Splice site mutations, which can affect normal splicing and gene expression, were most prevalent in JAK1 (37.5%), further emphasizing the potential impact of post-transcriptional modifications on pathway regulation. Notably, STAT3 and STAT5A/B exhibited a mix of missense, frame shift, and splice site mutations, indicating that multiple mutation mechanisms may be influencing the dysregulation of STAT signaling in this patient cohort. These findings provide a comprehensive characterization of JAK/STAT pathway mutations in H/L EOCRC patients, underscoring the predominance of missense mutations alongside structural alterations that may influence tumor progression and treatment response. The diversity of mutation types underscores the importance of functional validation studies to evaluate their role in colorectal cancer disparities and identify potential therapeutic targets for precision medicine strategies in H/L populations.

Review of MAPK pathway alterations in H/L LOCRC patients revealed missense mutations as the predominant alteration type, affecting key genes such as AKT1, BRAF, EGFR, KRAS, MAPK3, FGFR1-4, and NTRK2 (Appendix A). Frame shift deletions were observed in AKT2, CACNA1A, MAP2K4, MAP2K7, and TP53, while nonsense mutations were detected in AKT3, CACNG3, FGF13, MYC, and PDGFRA, indicating potential loss-of-function effects. Additionally, splicing mutations in NF1, RASA1, and TP53, along with translation start site mutations in FAS, suggest possible disruptions in protein synthesis and pathway regulation. These results point out distinct mutation patterns in the MAPK pathway among H/L LOCRC patients, emphasizing the need for further investigation into their functional and therapeutic implications.

Evaluation of JAK/STAT pathway alterations in H/L LOCRC patients revealed missense mutations as the most common alteration type, particularly in JAK2 (50%), JAK3 (50%), STAT3 (50%), STAT5A (66.7%), and STAT5B (33.3%) (Appendix A). Frame shift deletions were frequent in JAK1 (62.5%), JAK3 (33.3%), and STAT3 (50%), indicating potential loss-of-function effects. Additionally, frame shift insertions were observed in JAK1 (25%) and STAT5B (66.7%), while in-frame deletions were detected in JAK3 (16.7%) and STAT5A (16.7%), suggesting possible protein structure disruptions. Notably, nonsense mutations, which can lead to premature stop codons, were found in JAK2 (50%), indicating potential functional consequences.

Exploration of MAPK pathway alterations in NHW EOCRC patients revealed missense mutations as the predominant alteration type, affecting key genes such as AKT1, AKT2, BRAF, EGFR, FGFR1, KRAS, and MAP2K1 (Appendix A). Frame shift deletions in AKT1, BRAF, CDC25B, MAP3K1, and TP53 and nonsense mutations in ACVR1C, AKT3, BRAF, FGFR2, MAP2K4, and TP53 suggest potential loss-of-function effects. Additionally, in-frame deletions/insertions and splice site mutations in genes such as MAPK1, MYC, NF1, PDGFRA, and TGFBR2 may impact protein function and gene regulation.

Analysis of JAK/STAT pathway alterations in NHW EOCRC patients revealed missense mutations as the most frequent alteration type, particularly in JAK1, JAK2, JAK3, PIAS1, STAT1, and STAT3 (Appendix A). Frame shift deletions in JAK1, JAK2, JAK3, PTPRC, STAT3, and STAT5B, along with nonsense mutations in JAK2, PIAS2, and STAT3, suggest potential loss-of-function effects. Additionally, frame shift insertions in JAK1, JAK2, SOCS1, and STAT5A, and splice site mutations in JAK1, JAK3, and STAT3 indicate possible disruptions in gene regulation.

## 4. Discussion

The MAPK and JAK/STAT signaling pathways are fundamental regulators of cellular proliferation, differentiation, and survival, playing pivotal roles in CRC pathogenesis [18,19]. The activation of the MAPK pathway is a frequent occurrence in CRC, drives tumorigenesis by promoting cell proliferation and resistance to apoptosis, while the JAK/STAT pathway is essential in mediating immune signaling and inflammatory responses in tumor progression [21,22]. Although these pathways have been extensively studied in CRC, their specific roles and ethnic variations in EOCRC remain underexplored, particularly among H/L patients.

Our study provides a comprehensive analysis of MAPK and JAK/STAT pathway alterations in EOCRC, highlighting significant differences in mutation frequencies between H/L and NHW patients. Notably, key MAPK pathway genes such as NF1 (11.6% vs. 6.1%, *p* = 0.02), MAPK3 (3.6% vs. 0.7%, *p* = 6.83 × 10^−3^), AKT1 (5.1% vs. 1.8%, *p* = 0.03), and PDGFRB (5.8% vs. 2.1%, *p* = 0.02) were more frequently altered in H/L EOCRC patients compared to their NHW counterparts. These findings suggest that ethnicity-specific genetic variations in MAPK signaling may contribute to CRC disparities, potentially influencing tumor progression and response to targeted therapies [18,19,20].

Interestingly, when comparing EOCRC and LOCRC within the H/L cohort, significant differences were observed in MAPK-related gene alterations. NF1, ACVR1, and MAP2K1 mutations were significantly more prevalent in EOCRC, while BRAF mutations were markedly higher in LOCRC patients. These findings suggest that MAPK dysregulation may play a unique role in EOCRC among H/L patients, possibly contributing to the aggressive nature of CRC in this population. However, JAK/STAT pathway alterations did not show significant differences between EOCRC and LOCRC, nor between H/L and NHW patients, indicating that this pathway [21,22] may not be a major driver of ethnic disparities in CRC.

Our results align with previous studies that have demonstrated the role of the MAPK pathway in CRC progression and therapy resistance. Oncogenic activation of MAPK signaling, particularly via RAS and BRAF mutations, is known to promote aggressive tumor behavior and confer resistance to EGFR-targeted therapies in CRC patients [20]. The observed differences in MAPK pathway alterations between H/L and NHW EOCRC patients may indicate distinct tumor biology, potentially influencing treatment response and survival outcomes. Further research is needed to determine whether H/L patients exhibit differential responses to MAPK-targeted therapies, such as MEK and RAF inhibitors [19,20].

The survival analysis revealed no significant differences in overall survival for H/L EOCRC patients with or without MAPK pathway alterations (*p* = 0.9). This suggests that while MAPK alterations are prevalent in this population, their prognostic impact may be limited. In contrast, JAK/STAT pathway alterations in H/L EOCRC patients showed a trend toward worse survival outcomes, though the difference was only borderline significant (*p* = 0.073). These findings suggest a potential, albeit modest, prognostic role for JAK/STAT pathway alterations in EOCRC among H/L patients. In NHW EOCRC patients, JAK/STAT alterations were strongly associated with poorer survival outcomes (*p* < 0.0001), emphasizing their potential prognostic relevance in this ethnic group.

The observed ethnic disparities in MAPK and JAK/STAT pathway alterations may be influenced by genetic ancestry, environmental factors, and lifestyle differences. Studies have shown that genetic ancestry plays a significant role in cancer susceptibility and tumor biology, with H/L individuals exhibiting unique mutational profiles in various cancers, including CRC [6,7,8,9,10,22,23]. Additionally, dietary and lifestyle factors, such as higher consumption of processed foods and reduced access to preventive healthcare, may contribute to CRC risk and progression in this population [9]. Future research integrating genetic, environmental, and clinical data will be essential to elucidate the complex interactions driving CRC disparities.

The implications of our findings for precision medicine are significant. Given the high prevalence of MAPK pathway alterations in H/L EOCRC patients, therapies targeting this pathway, such as MEK inhibitors, may hold promise for this population [16,17]. Additionally, the potential prognostic impact of JAK/STAT pathway alterations justify further investigation into the efficacy of JAK inhibitors in CRC treatment. The lack of significant survival differences associated with MAPK alterations suggests that additional biomarkers, such as tumor microenvironment characteristics and immune signatures, should be explored to refine prognostic models and therapeutic strategies for H/L CRC patients.

Despite the strengths of this study, including the use of large-scale genomic data and rigorous statistical analyses, certain limitations must be taken into account. First, the analytical review-based approach in bioinformatics may introduce selection bias, as open-access genomic databases may not fully represent the broader H/L CRC patient group. Second, the relatively small sample size of H/L EOCRC patients may limit the statistical power to detect subtle differences in mutation frequencies and survival outcomes. Third, the lack of functional validation studies prevents direct mechanistic insights into how MAPK and JAK/STAT alterations contribute to CRC disparities.

This study specifically focuses on genetic mutations rather than gene expression changes. We examined mutation frequencies in MAPK and JAK/STAT pathway-related genes in EOCRC patients using publicly available genomic datasets. However, gene expression alterations were not included in our analysis, which also play a crucial role in cancer progression. While genetic mutations are a key aspect of tumor biology, future studies could enhance this research by incorporating gene expression changes to provide a more comprehensive understanding of the molecular mechanisms underlying EOCRC.

While our study specifically investigates genetic mutations in the MAPK and JAK/STAT pathways, we acknowledge that alterations in other pathways, such as the PI3K/AKT/mTOR pathway [39], also play critical roles in colorectal cancer progression. AKT1, though included in the MAPK pathway in our analysis, is primarily activated by the PI3K pathway and is an important factor in cell survival. However, alterations within the PI3K/AKT/mTOR pathway were not examined in this study. Given the relevance of this pathway in cancer biology, future studies should consider including PI3K/AKT/mTOR alterations to provide a more comprehensive understanding of the molecular mechanisms driving EOCRC, particularly among ethnic populations such as H/L patients.

To address these limitations, upcoming research should implement prospective cohort studies that encompass a broader and more heterogeneous H/L CRC patient cohort. Additionally, integrating multi-omics approaches, such as transcriptomics and proteomics, will provide a deeper understanding of the functional consequences of MAPK and JAK/STAT pathway alterations [40,41] in EOCRC. Investigating potential interactions between these pathways and other molecular drivers, such as WNT and PI3K signaling [7], will be crucial for developing targeted therapies tailored to H/L CRC patients.

The novelty of this study lies in its focused exploration of the MAPK and JAK/STAT pathways in EOCRC among H/L populations, a group that is often underrepresented in clinical and genomic research. Unlike previous studies, which have primarily concentrated on well-established pathways such as WNT and TGF-Beta, this research investigates the molecular alterations within the MAPK and JAK/STAT pathways, which play critical roles in tumor progression, immune response, and therapy resistance. The study is unique not only in its focus on these underexplored pathways in the context of H/L populations but also in its significant sample size of 302 H/L EOCRC patients, a number far larger than typically seen in studies of this population. This comprehensive approach, coupled with the focus on genetic mutations rather than gene expression changes, fills an important gap in understanding the molecular mechanisms driving colorectal cancer disparities. Furthermore, this study addresses a crucial public health concern, especially in regions with large, diverse populations, where the need for such research is particularly pressing. Ultimately, the findings presented here not only provide novel insights into the genetic underpinnings of EOCRC but also open the door for future therapeutic strategies tailored to the unique biology of underrepresented populations.

Comparison of MAPK and JAK/STAT pathway alterations in EOCRC H/L, LOCRC H/L, and EOCRC NHW patients revealed distinct mutation patterns that may contribute to differences in tumor biology and treatment response. In EOCRC H/L patients, missense mutations were the predominant alteration type in both pathways, with frame shift deletions, nonsense mutations, and splice site alterations occurring at lower frequencies. This suggests that single amino acid substitutions play a major role in MAPK and JAK/STAT pathway dysregulation in this population. In contrast, LOCRC H/L patients exhibited a greater presence of structural mutations, including frame shift deletions, nonsense mutations, and splice site mutations, particularly in TP53, NF1, and RASA1, which could indicate more profound disruptions in protein function and pathway regulation. The presence of translation start site mutations in FAS further highlights potential post-transcriptional alterations unique to this group. These findings suggest distinct molecular mechanisms may be driving tumor progression in LOCRC compared to EOCRC within the H/L population. Among EOCRC NHW patients, missense mutations were also the most common alteration type, affecting genes such as BRAF, AKT1, EGFR, and KRAS, similar to EOCRC H/L patients. However, EOCRC NHW patients displayed a higher proportion of frame shift deletions, in-frame deletions, and nonsense mutations, particularly in MAP3K1, MAP2K4, and TP53, suggesting a greater likelihood of loss-of-function effects. Additionally, splice site mutations in NF1, PDGFRA, and TGFBR2 highlight potential gene regulation disruptions in this population. These findings suggest that while missense mutations are a common feature in both EOCRC H/L and NHW patients, structural mutations such as frame shift deletions and nonsense mutations are more prominent in LOCRC H/L and EOCRC NHW patients, potentially influencing tumor behavior, response to therapy, and disease progression. The variability in mutation types across these groups underscores the importance of ethnicity- and age-specific analyses in colorectal cancer research, highlighting the need for precision medicine approaches that account for these molecular differences. Further functional studies are necessary to determine the clinical impact of these mutation patterns and their potential as therapeutic targets in H/L and NHW CRC patients.

A consideration in the interpretation of our findings is the distinction between somatic mutations and germline polymorphisms, as both can be present in publicly available datasets. Our analysis focused on somatic mutations relevant to tumorigenesis, as reported in cancer-specific databases. However, some variants may represent germline polymorphisms rather than oncogenic drivers, which could influence the observed mutation patterns. While our study identified ethnicity-specific differences in MAPK and JAK/STAT pathway alterations, further functional validation is necessary to confirm the oncogenic significance of these alterations. Additionally, we acknowledge that the absence of comprehensive germline vs. somatic classification in certain datasets presents a limitation, which should be addressed in future studies through deep sequencing and functional assays. These insights underscore the need for precision medicine approaches that account for both genetic background and tumor-specific alterations in understanding colorectal cancer disparities.

While publicly available datasets are valuable for large-scale analyses, a key limitation of our study is their lack of comprehensive functional annotations, making it challenging to differentiate oncogenic mutations from benign polymorphisms. Future studies should incorporate deep sequencing techniques, functional assays, and experimental validation to determine the precise impact of these mutations on tumor progression, drug response, and patient outcomes. Additionally, integrating matched normal-tumor sequencing data could help further distinguish somatic driver mutations from inherited polymorphisms, improving the accuracy of molecular characterizations in colorectal cancer disparities research.

Although false discovery rate (FDR) correction is commonly used in genomic studies to control for multiple testing, our study reports *p*-values without FDR adjustment, which may present a potential limitation. Although FDR reduces the likelihood of false positives, it can also overcorrect in studies with smaller sample sizes, potentially excluding biologically significant findings [42]. Our approach aligns with previous studies [37,38,43] that have prioritized biological relevance over strict statistical significance, particularly when analyzing mutation prevalence and pathway alterations in ethnicity-specific colorectal cancer cohorts. Given the exploratory nature of our study, we aimed to capture patterns of mutation enrichment that may be relevant to tumor progression and treatment response rather than imposing stringent statistical cutoffs that could obscure clinically significant insights. However, we acknowledge that the absence of FDR correction may increase the risk of false positives, and future studies with larger cohorts should incorporate both FDR-adjusted and unadjusted analyses to balance statistical rigor and biological interpretability.

This study highlights significant ethnicity-specific differences in MAPK pathway alterations in EOCRC, with H/L patients exhibiting a higher prevalence of key mutations compared to NHW patients. While JAK/STAT pathway alterations were not significantly different between ethnic groups, their potential prognostic impact in H/L patients justify further investigation. These findings underscore the importance of precision medicine approaches that account for genetic and ethnic heterogeneity in CRC, with the goal of improving treatment outcomes and reducing cancer disparities in underrepresented populations.

## 5. Conclusions

In conclusion, our findings provide new insights into the ethnicity-specific molecular alterations in the MAPK and JAK/STAT pathways in EOCRC, particularly among H/L patients. The significantly higher prevalence of key MAPK pathway alterations, including NF1, ACVR1, and MAP2K1, in H/L EOCRC patients suggests a potential role for MAPK dysregulation in CRC disparities. Additionally, while JAK/STAT pathway alterations were not significantly different between H/L and NHW patients, their borderline association with survival outcomes in H/L EOCRC patients justify further investigation. These findings underscore the need for further research into the molecular drivers of EOCRC disparities, as well as the potential clinical implications of MAPK and JAK/STAT pathway alterations. The observed ethnicity-specific differences highlight the importance of precision medicine approaches tailored to diverse populations. Future studies should integrate multi-omics analyses and clinical outcomes to refine targeted therapies that address the unique tumor biology of H/L CRC patients. By advancing our understanding of the genetic landscape of EOCRC, these insights may contribute to reducing cancer disparities and improving treatment strategies for underrepresented populations.

## Figures and Tables

**Figure 1 cancers-17-01093-f001:**
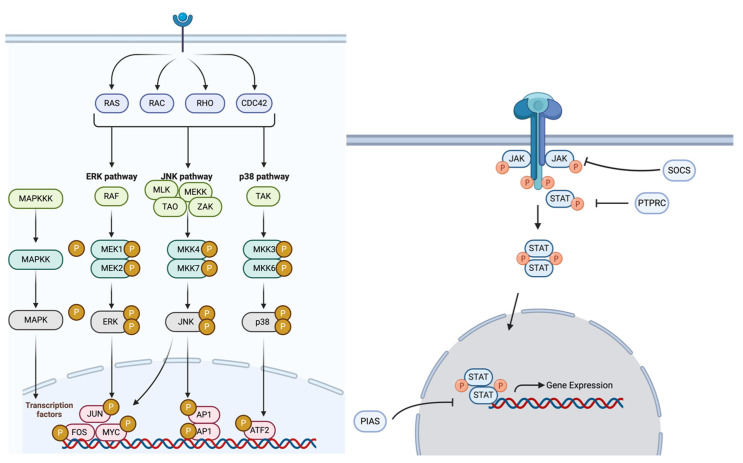
Illustration of the MAPK (**left**) and JAK/STAT (**right**) signaling pathways. In the left panel, the MAPK signaling pathway is shown, emphasizing its critical role in controlling cell proliferation, differentiation, survival, and response to stress. This pathway is activated through various receptor-mediated signals, including RAS, RAC, RHO, and CDC42, leading to three major downstream cascades: ERK, JNK, and p38 pathways. While the ERK pathway predominantly regulates cell growth and specialization, the JNK and p38 pathways play key roles in managing cellular stress, programmed cell death, and immune regulation. Activation of these pathways results in the phosphorylation and activation of transcription factors such as JUN, FOS, MYC, AP1, and ATF2, which influence gene expression and tumor progression. Dysregulation of MAPK pathway genes, including NF1, BRAF, AKT1, and MAPK3, has been implicated in early-onset colorectal cancer (EOCRC), particularly among H/L patients. The right panel illustrates the JAK/STAT (Janus kinase/signal transducer and activator of transcription) pathway, a key intracellular signaling cascade involved in immune regulation, inflammation, and tumorigenesis. Ligand binding to cytokine receptors induces the phosphorylation of JAK kinases, which subsequently activate STAT proteins through phosphorylation. Phosphorylated STAT dimers translocate to the nucleus, where they regulate gene expression involved in cell proliferation, survival, and immune responses. This pathway is negatively regulated by SOCS and PTPN6, which act as feedback inhibitors.

**Figure 2 cancers-17-01093-f002:**
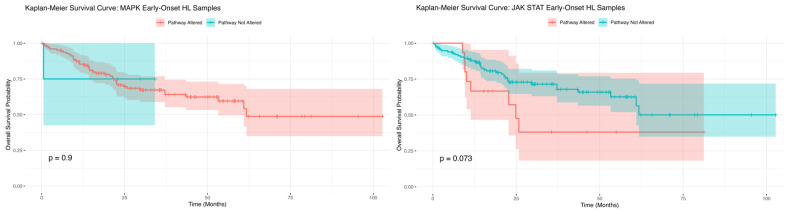
Overall survival curves of early-onset colorectal cancer (EOCRC) Hispanic/Latino (H/L) patients stratified by the presence or absence of MAPK (**left**) and JAK/STAT (**right**) pathway alterations. The left panel illustrates the Kaplan–Meier survival curve for EOCRC H/L patients stratified by MAPK pathway alterations. Patients with MAPK pathway alterations (red curve) exhibited no significant survival difference compared to those without alterations (blue curve) (*p* = 0.9). The shaded regions around the curves represent 95% confidence intervals, and vertical tick marks indicate censored patients. These results suggest that MAPK pathway alterations may not have a strong prognostic impact on overall survival in EOCRC H/L patients. The right panel presents the Kaplan–Meier survival curve for EOCRC H/L patients stratified by JAK/STAT pathway alterations. Patients with JAK/STAT pathway alterations (red curve) showed a trend toward worse survival outcomes compared to those without alterations (blue curve), though this difference was only borderline significant (*p* = 0.073).

**Table 1 cancers-17-01093-t001:** Demographic and clinical characteristics of Hispanic/Latino (H/L) and Non-Hispanic White (NHW) cohorts.

Clinical Characteristics	H/L Cohort *n* (%)	NHW Cohort *n* (%)
Age Onset and Gender		
Early-Onset (<50 y) Female	55 (18.2%)	394 (12.7%)
Early-Onset (<50 y) Male	83 (27.5%	503 (16.2%)
Late-Onset (≥50 y) Female	71 (23.5%)	1004 (32.3%)
Late-Onset (≥50 y) Male	93 (30.8%)	1209 (38.9%)
Stage at Diagnosis		
Stage 1–3	98 (32.5%)	965 (31.0%)
Stage 4	132 (43.7%)	1131 (36.4%)
NA	72 (23.8%)	1014 (32.6%)
Ethnicity		
Spanish NOS; Hispanic NOS, Latino NOS	270 (89.4%)	0 (0.0%)
Mexican (includes Chicano)	28 (9.3%)	0 (0.0%)
Other Spanish/Hispanic	1 (0.3%)	0 (0.0%)
Spanish surname only	3 (1.0%)	0 (0.0%)
Non-Spanish; Non-Hispanic	0 (0.0%)	3110 (100.0%)

**Table 2 cancers-17-01093-t002:** Ethnicity-associated differences in clinical characteristics between Hispanic/Latino (H/L) and Non-Hispanic White (NHW) cohorts.

Clinical Feature	Early-Onset HL *n* (%)	Late-Onset HL *n* (%)	*p*-Value	Early-Onset HL *n* (%)	Early-Onset NHW *n* (%)	*p*-Value
Median Diagnosis Age (IQR)	41 (36–46)	61 (55–69)	<0.05	41 (36–46)	42 (38–47)	<0.05
Median Mutation Count *	7 (5–10)	8 (6–10)	>0.05	7 (5–10)	7 (5–10)	>0.05
ACVR1 Mutation						
Present	4 (2.9%)	0 (0.0%)	<0.05	4 (2.9%)	11 (1.2%)	>0.05
Absent	134 (97.1%)	164 (100.0%)		134 (97.1%)	886 (98.8%)	
AKT1 Mutation						
Present	7 (5.1%)	3 (1.8%)	>0.05	7 (5.1%)	16 (1.8%)	<0.05
Absent	131 (94.9%)	161 (98.2%)		131 (94.9%)	881 (98.2%)	
BRAF Mutation						
Present	7 (5.1%)	30 (18.3%)	<0.05	7 (5.1%)	67 (7.5%)	>0.05
Absent	131 (94.9%)	134 (81.7%)		131 (94.9%)	830 (92.5%)	
MAPK2K1 Mutation						
Present	5 (3.6%)	0 (0.0%)	<0.05	5 (3.6%)	16 (1.8%)	>0.05
Absent	133 (96.4%)	164 (100.0%)		133 (96.4%)	881 (98.2%)	
MAPK3 Mutation						
Present	5 (3.6%)	1 (0.6%)	>0.05	5 (3.6%)	6 (0.7%)	<0.05
Absent	133 (96.4%)	163 (99.4%)		133 (96.4%)	891 (99.3%)	
NF1 Mutation						
Present	16 (11.6%)	6 (3.7%)	<0.05	16 (11.6%)	55 (6.1%)	<0.05
Absent	122 (88.4%)	158 (96.3%)		122 (88.4%)	842 (93.9%)	
PDGFRB Mutation						
Present	8 (5.8%)	3 (1.8%)	>0.05	8 (5.8%)	19 (2.1%)	<0.05
Absent	130 (94.2%)	161 (98.2%)		130 (94.2%)	878 (97.9%)	

* LO HL: NA 1, EO NHW: NA 8.

**Table 3 cancers-17-01093-t003:** Frequency of MAPK and JAK/STAT pathway alterations in early-onset (EOCRC) and late-onset (LOCRC) colorectal cancer among Hispanic/Latino (H/L) patients.

	Early-Onset H/L *n* (%)	Late-Onset H/L *n* (%)	*p*-Value
JAK/STAT Alterations Present	16 (11.6%)	19 (11.6%)	1
JAK/STAT Alterations Absent	122 (88.4%)	145 (88.4%)	
MAPK Alterations Present	134 (97.1%)	161 (98.2%)	0.7063
MAPK Alterations Absent	4 (2.9%)	3 (1.8%)	

**Table 4 cancers-17-01093-t004:** Frequency of MAPK and JAK/STAT pathway alterations in early-onset colorectal cancer (EOCRC) among Hispanic/Latino (H/L) and Non-Hispanic White (NHW) patients.

	Early Onset H/L *n* (%)	Early Onset NHW *n* (%)	*p*-Value
JAK/STAT Alterations Present	16 (11.6%)	78 (8.7%)	0.3452
JAK/STAT Alterations Absent	122 (88.4%)	819 (91.3%)	
MAPK Alterations Present	137 (99.3%)	861 (96.0%)	0.6604
MAPK Alterations Absent	4 (2.9%)	36 (4.0%)	

## Data Availability

All data used in the present study are publicly available at https://www.cbioportal.org/ and https://genie.cbioportal.org. Additional data can be provided upon reasonable request to the authors.

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
