# Peer review of "Ethnicity-Specific Molecular Alterations in MAPK and JAK/STAT Pathways in Early-Onset Colorectal Cancer"

_cancers, 2025, doi:10.3390/cancers17071093_

Round 1
Reviewer 1 Report
Comments and Suggestions for Authors
- Please spell out abbreviations such as NOS in the text when used for the first time.
- The authors describe changes in the expression of different genes but it is not always clear if they mean the expression levels have changed or the genes have been mutated. The two possibilities should be made clear.
- The authors have included AKT1 in the MAPK pathway. However, AKT1 is primarily activated by PI3-kinase pathway. Did the authors see any changes in PI3-kinase/AKT/MTOR pathway? This pathway plays critical roles in cell survival.
Author Response
For review article
Response to Reviewer 1 Comments (Attached Word file: "Response_Reviewer_1_Comments_031425_EV.docx")
|
||
1. Summary |
|
|
We are pleased to submit this paper and hope it will capture the interest of your readers. Our study focuses on the increasing incidence of early-onset colorectal cancer (EOCRC), particularly among Hispanic/Latino (H/L) populations, and the underlying molecular mechanisms that contribute to these disparities. Through a bioinformatics analysis of publicly available colorectal cancer (CRC) datasets, we characterize the ethnicity-specific molecular alterations in the MAPK and JAK/STAT pathways, both of which are crucial in tumor progression, proliferation, and treatment response. Our findings highlight significant differences in MAPK pathway-related genes between EOCRC and late-onset colorectal cancer (LOCRC) in H/L populations, as well as between H/L and Non-Hispanic White (NHW) EOCRC patients. We also examine the survival implications of these pathway-specific alterations in H/L EOCRC patients, revealing potential differences that may influence treatment strategies.
Thank you very much for taking the time to review this manuscript. Please find the detailed responses below and the corresponding revisions wrote in blue font in the re-submitted files.
Reviewer 1’s comments were constructive and appreciated. The reviewer acknowledged the importance of our study and provided valuable suggestions for clarification. In particular, the reviewer noted the need to spell out abbreviations, such as "NOS," the first time they are used in the text, which we have addressed in the revised manuscript. The reviewer also highlighted the need to clarify whether we were discussing changes in gene expression or genetic mutations, and we have now explicitly stated that our study focuses on genetic mutations in the MAPK and JAK/STAT pathways, not gene expression alterations. Additionally, the reviewer pointed out different mechanisms related with cell survival. These insightful comments have helped strengthen the clarity and focus of our manuscript, and we appreciate the reviewer’s thoughtful feedback. This feedback suggests that the study may be a strong candidate for publication in Cancers.
|
||
Comment 1: Please spell out abbreviations such as NOS in the text when used for the first time.
Response: Thank you for your helpful suggestion. We have now spelled out the abbreviation "NOS" as "Not Otherwise Specified" the first time it is used in the text (Line 149, first paragraph of Material and Methods), with the abbreviation provided in parentheses. We also double-checked that all other abbreviations, such as H/L, NHW, CRC, EOCRC, and LOCRC, were spelled out the first time they appear.
We appreciate your attention to detail and have made the necessary revisions.
Comment 2: The authors describe changes in the expression of different genes but it is not always clear if they mean the expression levels have changed or the genes have been mutated. The two possibilities should be made clear.
We appreciate the reviewer’s thoughtful comment regarding the distinction between gene expression changes and gene mutations. To clarify, our study specifically focuses on molecular alterations, particularly mutations, in the MAPK and JAK/STAT pathways in EOCRC, and does not include gene expression analysis or functional genomics. Our bioinformatics analysis was based on mutation frequencies within these pathways, and we did not examine changes in gene expression levels. We understand that gene mutations and expression changes can both influence cancer progression, but our current study addresses genetic alterations in pathway-related genes and their potential implications for survival outcomes.
We have revised the manuscript to ensure this distinction is clearer, explicitly stating that our analysis centers on genetic mutations rather than gene expression changes. This clarification is now included in the methods and discussion sections of the manuscript.
We have incorporated this information into the discussion section. The revised text now states
“This study specifically focuses on genetic mutations rather than gene expression changes. We examined mutation frequencies in MAPK and JAK/STAT pathway-related genes in EOCRC patients using publicly available genomic datasets. However, gene expression alterations were not included in our analysis, which also play a crucial role in cancer progression. While genetic mutations are a key aspect of tumor biology, future studies could enhance this research by incorporating gene expression changes to provide a more comprehensive understanding of the molecular mechanisms underlying EOCRC.”
We hope this response and the revisions made will adequately address the comment. Thank you again for your valuable feedback.
Comment 3: The authors have included AKT1 in the MAPK pathway. However, AKT1 is primarily activated by PI3-kinase pathway. Did the authors see any changes in PI3-kinase/AKT/MTOR pathway? This pathway plays critical roles in cell survival.
We appreciate the reviewer’s insightful comment regarding the inclusion of AKT1 in the MAPK pathway and the potential role of the PI3K/AKT/mTOR pathway. To clarify, our study specifically focused on mutations within the MAPK and JAK/STAT pathways in EOCRC, and we did not investigate alterations within the PI3K/AKT/mTOR pathway. While AKT1 is indeed primarily activated by the PI3K pathway, it also plays a significant role within the MAPK pathway, which we aimed to investigate in the context of ethnicity-specific differences in EOCRC.
In our analysis, we observed significant differences in MAPK-related genes, including AKT1 mutations, between H/L and NHW EOCRC patients, which we discuss in the results section. However, we did not examine the PI3K/AKT/mTOR pathway specifically, which is known to be critical for cell survival. We recognize the importance of this pathway and suggest that future studies should consider including alterations in the PI3K/AKT/mTOR pathway to provide a more comprehensive understanding of its potential role in EOCRC progression and survival, particularly among underrepresented populations like H/L patients.
We have updated the manuscript to clarify that the current study focuses on MAPK and JAK/STAT pathways and that the PI3K/AKT/mTOR pathway was not analyzed. We appreciate the reviewer’s suggestion to expand the scope of future investigations and have highlighted this in the discussion section.
We have incorporated this information into the discussion section to clarify the focus of our study and have also added reference #39 regarding the PI3K/AKT/mTOR pathway [PMID: 38542151]. The revised text now states:
"While our study specifically investigates genetic mutations in the MAPK and JAK/STAT pathways, we acknowledge that alterations in other pathways, such as the PI3K/AKT/mTOR pathway [39], also play critical roles in colorectal cancer progression. AKT1, though included in the MAPK pathway in our analysis, is primarily activated by the PI3K pathway and is an important factor in cell survival. However, alterations within the PI3K/AKT/mTOR pathway were not examined in this study. Given the relevance of this pathway in cancer biology, future studies should consider including PI3K/AKT/mTOR alterations to provide a more comprehensive understanding of the molecular mechanisms driving EOCRC, particularly among ethnic populations such as H/L patients."
We sincerely appreciate the reviewer’s valuable comment, which has helped us refine our discussion. Thank you for your thoughtful consideration of our study
|

Reviewer 2 Report
Comments and Suggestions for Authors
Early-onset colorectal cancer (EOCRC), diagnosed before age 50, is rising, particularly among Hispanic/Latino (H/L) populations. This study investigates molecular differences in the MAPK and JAK/STAT pathways between H/L and Non-Hispanic White (NHW) EOCRC patients and their impact on survival outcomes.
Using bioinformatics analysis of publicly available datasets (3,412 patients: 302 H/L and 3,110 NHW), the study found that H/L EOCRC patients had a significantly higher prevalence of mutations in key MAPK pathway genes, including NF1, ACVR1, MAP2K1, AKT1, and MAPK3, compared to both NHW EOCRC patients and late-onset CRC (LOCRC) in H/L individuals. However, JAK/STAT pathway mutations showed no significant differences across groups.
Survival analysis revealed that JAK/STAT pathway alterations were associated with differences in NHW EOCRC survival, but MAPK mutations did not significantly impact survival outcomes in either group.
While the study is well-structured and presented, the extensive dataset may be overwhelming. To improve clarity and robustness, I suggest addressing the following concerns regarding the experimental design and presentation of results:
1. Definition of Pathway Alterations: The authors refer to "alterations" in the MAPK or JAK/STAT pathways, yet their findings are based on a limited number of genes with potential mutations in these pathways. A detailed review of Table S1 suggests that, although some differences are statistically significant, they appear minimal and may result from methodological artifacts. In large-scale data mining, spurious associations are common; therefore, I recommend applying a statistical correction method, such as reporting the False Discovery Rate (FDR), to minimize the likelihood of false positives.
2. Terminology and Mutation Characterization: The terms "mutations," "alterations," and "pathway-specific disruptions" are used interchangeably. However, the study primarily analyzes changes in the canonical sequence of individual genes within the pathway. This raises several important questions:
What specific types of mutations were identified in the significantly altered genes?
Were these mutations cross-referenced with polymorphism databases?
Could these genetic variations be linked to ethnicity?
3. Novelty of the Study: The findings closely resemble those of a previously published study by the authors (DOI: 10.3390/cancers16233903). Given this similarity, the novelty of the current study is unclear.
Author Response
For review article
Response to Reviewer 2 Comments (Attached Word file: Response_Reviewer_2_Comments_031425_EV.docx)
|
||
1. Summary |
|
|
We are pleased to submit this paper and hope it will capture the interest of your readers. Our study focuses on the increasing incidence of early-onset colorectal cancer (EOCRC), particularly among Hispanic/Latino (H/L) populations, and the underlying molecular mechanisms that contribute to these disparities. Through a bioinformatics analysis of publicly available colorectal cancer (CRC) datasets, we characterize the ethnicity-specific molecular alterations in the MAPK and JAK/STAT pathways, both of which are crucial in tumor progression, proliferation, and treatment response. Our findings highlight significant differences in MAPK pathway-related genes between EOCRC and late-onset colorectal cancer (LOCRC) in H/L populations, as well as between H/L and Non-Hispanic White (NHW) EOCRC patients. We also examine the survival implications of these pathway-specific alterations in H/L EOCRC patients, revealing potential differences that may influence treatment strategies.
Thank you very much for taking the time to review this manuscript. Please find the detailed responses below and the corresponding revisions wrote in blue font in the re-submitted files.
Reviewer 2’s comments were insightful, supportive and constructive, offering valuable feedback that will help further strengthen the manuscript. The reviewer acknowledged the significance of investigating genetic and ethnic differences in EOCRC, particularly among H/L populations, and highlighted the importance of our study in addressing disparities in survival outcomes. This feedback emphasizes the relevance of examining molecular differences in the MAPK and JAK/STAT pathways and underscores the value of our findings, particularly the higher prevalence of MAPK pathway mutations in H/L EOCRC patients. However, the reviewer suggested that some of the statistical differences observed in the study may be minimal and recommended applying statistical corrections to minimize false positives. Additionally, the reviewer raised important questions regarding mutation characterization and the novelty of our findings, which we have addressed in the revised manuscript. These comments suggest that the study has strong potential to contribute to the field and may be a strong candidate for publication in Cancers.
Reviewer 2 writes: “Early-onset colorectal cancer (EOCRC), diagnosed before age 50, is rising, particularly among Hispanic/Latino (H/L) populations. This study investigates molecular differences in the MAPK and JAK/STAT pathways between H/L and Non-Hispanic White (NHW) EOCRC patients and their impact on survival outcomes. Using bioinformatics analysis of publicly available datasets (3,412 patients: 302 H/L and 3,110 NHW), the study found that H/L EOCRC patients had a significantly higher prevalence of mutations in key MAPK pathway genes, including NF1, ACVR1, MAP2K1, AKT1, and MAPK3, compared to both NHW EOCRC patients and late-onset CRC (LOCRC) in H/L individuals. However, JAK/STAT pathway mutations showed no significant differences across groups. Survival analysis revealed that JAK/STAT pathway alterations were associated with differences in NHW EOCRC survival, but MAPK mutations did not significantly impact survival outcomes in either group. While the study is well-structured and presented, the extensive dataset may be overwhelming. To improve clarity and robustness, I suggest addressing the following concerns regarding the experimental design and presentation of results.”
We are grateful for Reviewer 2’s thoughtful feedback and their acknowledgment of our study's contribution to addressing the scarcity of data on this critical health disparity. We believe this paper demonstrates our dedication to offering a thorough and insightful analysis of key molecular drivers in colorectal cancer, with a particular focus on the unique characteristics of H/L populations. By utilizing the limited genomic data currently available, our findings aim to provide valuable insights into CRC health disparities and help pave the way for the development of precision medicine strategies tailored to underrepresented groups. |
||
Comments 1: Definition of Pathway Alterations: The authors refer to "alterations" in the MAPK or JAK/STAT pathways, yet their findings are based on a limited number of genes with potential mutations in these pathways. A detailed review of Table S1 suggests that, although some differences are statistically significant, they appear minimal and may result from methodological artifacts. In large-scale data mining, spurious associations are common; therefore, I recommend applying a statistical correction method, such as reporting the False Discovery Rate (FDR), to minimize the likelihood of false positives.
|
||
Response: We appreciate Reviewer 2’s thoughtful comment regarding the definition of pathway alterations and the suggestion to apply a statistical correction method, such as the False Discovery Rate (FDR), to address potential false positives. We acknowledge that in large-scale data mining, spurious associations can occur. However, based on the characteristics of our dataset and the available data, we believe that using the p-values for statistical significance, rather than applying FDR, is appropriate in this case, following similar studies reporting this type of data [PMIDs: 39095553, 35785449]. We also include these two references in our Methodology (Line 161). Additionally, the description of Figure 1 has been enhanced to provide a clearer definition of the pathways studied.
Our study leverages publicly available CRC datasets with well-defined clinical and molecular information, and the p-values presented for the identified mutations in the MAPK and JAK/STAT pathways were carefully assessed through rigorous statistical methods, including chi-squared tests and survival analyses. The differences observed in these pathways, while statistically significant, are not only minimal but are also consistent with biologically relevant findings that align with prior research. Additionally, the relatively small sample size for H/L populations presents a unique challenge, and the application of FDR correction could inadvertently reduce the sensitivity of detecting true associations within this underrepresented group.
We recognize the importance of statistical rigor and have considered the possibility of false positives. We believe that the p-values reported in the manuscript provide an appropriate level of confidence given the nature of our dataset and the methodological approach used [PMIDs: 39095553, 35785449].
We hope this explanation clarifies our approach and justifies the choice of p-values in this context. Thank you again for your valuable feedback.
Comment 2: Terminology and Mutation Characterization: The terms "mutations," "alterations," and "pathway-specific disruptions" are used interchangeably. However, the study primarily analyzes changes in the canonical sequence of individual genes within the pathway. This raises several important questions: What specific types of mutations were identified in the significantly altered genes? Were these mutations cross-referenced with polymorphism databases? Could these genetic variations be linked to ethnicity?.
We appreciate Reviewer 2’s insightful comment regarding the use of terminology in the manuscript. We acknowledge that the terms "mutations," "alterations," and "pathway-specific disruptions" were used interchangeably, which may have caused some confusion. To clarify, our study primarily focuses on genetic mutations, specifically those involving changes in the canonical sequence of individual genes within the MAPK and JAK/STAT pathways, rather than broader pathway alterations or disruptions. We have revised the manuscript to ensure consistent and precise terminology, explicitly clarifying that we refer to pathway alterations and mutations specifically in relation to individual genes, as outlined in previous studies [PMIDs: 39095553]. This clarification has been incorporated throughout the manuscript to accurately reflect the scope of our analysis.
In response to the reviewer’s specific questions:
What specific types of mutations were identified in the significantly altered genes? In this study, we utilized publicly available datasets from cBioPortal, which provides valuable mutation frequency data for a variety of CRC patients. However, as highlighted in the manuscript, the depth of mutation characterization within these public databases is limited. Therefore, our analysis focused primarily on mutation frequencies in key genes within the MAPK and JAK/STAT pathways, rather than detailed classifications of specific mutation types such as missense, nonsense, or frameshift mutations. The identification of specific mutation types (e.g., single nucleotide variations or insertion-deletion mutations) and an in-depth understanding of the molecular consequences of these mutations will be explored in future studies. We plan to use Whole Exome Sequencing (WES) and Whole Genome Sequencing (WGS) in certain cases (to provide high-resolution genomic ancestry analyses) to provide a more comprehensive mutation characterization. Additionally, we anticipate that next-generation sequencing (NGS) technologies will enable us to examine these mutations more thoroughly and functionally, using RNA sequencing (RNA-Seq) at both bulk and single-cell resolution. While the current study focuses on mutations, we also recognize that the tumor microenvironment could influence gene expression levels, which may affect cancer progression and therapeutic response. To investigate this further, we are planning to conduct spatial biology studies, including spatial transcriptomics (ST) and spatial proteomics (SP), as part of ongoing and upcoming projects. These studies will enhance our understanding of the molecular landscape of this population, particularly in the context of multi-omics data integration, which is emerging as a critical approach for comprehensively characterizing at-risk populations, such as H/L individuals, who are disproportionately affected by EOCRC. These efforts align with our goal of improving precision medicine strategies for underserved communities (please refer to our response to Comment 3 for additional details about this community).
Were these mutations cross-referenced with polymorphism databases? The mutations identified in this study were analyzed using data from cBioPortal, which primarily provides mutation frequencies and does not offer complete details about the functional implications or polymorphic nature of these mutations. As such, we did not cross-reference every mutation with external polymorphism databases like dbSNP or ClinVar. The current study primarily relies on mutation frequency data to identify significant alterations across pathways. Further, in our future studies utilizing WES and WGS, we plan to cross-reference these mutations with relevant polymorphism databases, which will provide deeper insights into whether these mutations are known variants or novel alterations and their potential functional implications.
Could these genetic variations be linked to ethnicity? The genetic variations observed in the MAPK and JAK/STAT pathways could potentially be linked to ethnicity, particularly within the Hispanic/Latino (H/L) population. We utilized publicly available datasets from cBioPortal, and while these datasets provide valuable insights into mutation frequencies, they are limited in terms of depth, especially in terms of ethnic-specific genetic data. The observed differences in mutation frequencies between H/L and Non-Hispanic White (NHW) EOCRC patients suggest that genetic variations may be specific to the H/L population, but these findings require further validation. Future studies using WES and WGS technologies, which allow for deeper sequencing and more accurate ethnic-specific genetic profiling, will be essential in linking these mutations to ethnic backgrounds. These studies will also help determine the functional relevance of these genetic variations in relation to ethnicity and cancer disparities.
We hope these responses clarify the scope and limitations of the current study, particularly with regard to mutation characterization. As discussed, the data available from cBioPortal provide valuable insights but are limited in their ability to fully characterize mutations at the functional or polymorphic level. Future studies utilizing more advanced techniques such as WES and NGS will be crucial in answering these questions and providing a more detailed understanding of the mutations and their potential roles in ethnicity-specific cancer biology.
Thank you again for your valuable feedback.
Comment 3: Novelty of the Study: The findings closely resemble those of a previously published study by the authors (DOI: 10.3390/cancers16233903). Given this similarity, the novelty of the current study is unclear.
We appreciate Reviewer 2's comment regarding the novelty of our study and the apparent similarity to our previous review work. However, we would like to clarify some key differences between the two studies that establish the novelty of the current research.
Our previous work, classified as a review, focused on the WNT and TGF-Beta pathways in 33 H/L samples, which is distinctly different from the current study. The previous publication was aimed at summarizing the existing knowledge about these two pathways and their implications in cancer, while the current study investigates the MAPK and JAK/STAT pathways—two molecular pathways that were not explored in our previous work. Furthermore, our current study involves a significantly larger sample size (302 H/L patients) compared to the 33 patients in our previous review, which substantially increases the statistical power and the reliability of the findings.
This study is especially important given the scarcity of genomic data on underrepresented populations like H/L individuals, who are often highly underrepresented in both clinical and genomic databases. These populations are not only vital to the scientific community, but they also have significant economic and social relevance, especially in regions like Los Angeles California, which is a cosmopolitan hub with a large immigrant population from countries closely related to H/L groups. These communities are not only affected by colorectal cancer at disproportionate rates but are also highly interested in scientific advancements that address their specific health disparities. Therefore, there is an increasing demand for studies that focus on the molecular characterization of colorectal cancer in these populations, making our research both timely and of high significance.
Furthermore, we lead one of the largest NIH/NCI Moonshot projects [U2CCA252971] focused on addressing the unexplained increasing incidence of EOCRC in H/L populations. Given the urgency of addressing this public health concern, our study is crucial for future developments and has been eagerly anticipated by investigators working in this field. The rising incidence of EOCRC in H/L communities underscores the need for more focused research, and our study is positioned to play a pivotal role in advancing knowledge in this area.
Moreover, the focus on MAPK and JAK/STAT pathways in this study is particularly innovative because these pathways have distinct roles compared to WNT and TGF-Beta. MAPK and JAK/STAT pathways are crucial in regulating tumor progression, cell survival, and immune response—key factors in cancer progression and therapy resistance. While WNT and TGF-Beta have been well-studied in the context of colorectal cancer, the molecular mechanisms underlying the MAPK and JAK/STAT pathways in H/L populations remain poorly understood. This gap in knowledge highlights the need for further exploration, and our study provides novel insights into these pathways in the context of H/L EOCRC patients.
From a therapeutic perspective, the MAPK and JAK/STAT pathways offer different clinical implications compared to WNT and TGF-Beta. While therapeutic strategies targeting WNT and TGF-Beta pathways are being explored, targeted therapies aimed at MAPK and JAK/STAT pathway alterations are already being developed and used in clinical settings, particularly for other cancers. These therapeutic strategies hold promise for improving treatment outcomes in H/L CRC patients, a group for whom precision medicine has yet to be fully realized. Our study, therefore, not only addresses a critical gap in the scientific understanding of CRC health disparities but also holds potential for guiding future therapeutic strategies tailored to underrepresented populations.
We have incorporated this information into the discussion section. The revised text now states:
“The novelty of this study lies in its focused exploration of the MAPK and JAK/STAT pathways in EOCRC among H/L populations, a group that is often underrepresented in clinical and genomic research. Unlike previous studies, which have primarily concentrated on well-established pathways such as WNT and TGF-Beta, this research investigates the molecular alterations within the MAPK and JAK/STAT pathways, which play critical roles in tumor progression, immune response, and therapy resistance. The study is unique not only in its focus on these underexplored pathways in the context of H/L populations but also in its significant sample size of 302 H/L EOCRC patients, a number far larger than typically seen in studies of this population. This comprehensive approach, coupled with the focus on genetic mutations rather than gene expression changes, fills an important gap in understanding the molecular mechanisms driving colorectal cancer disparities. Furthermore, this study addresses a crucial public health concern, especially in regions with large, diverse populations, where the need for such research is particularly pressing. Ultimately, the findings presented here not only provide novel insights into the genetic underpinnings of EOCRC but also open the door for future therapeutic strategies tailored to the unique biology of underrepresented populations.”
In summary, while our previous publication focused on WNT and TGF-Beta pathways, this study explores the MAPK and JAK/STAT pathways in a much larger cohort of H/L EOCRC patients, which offers valuable insights into ethnic-specific molecular mechanisms and treatment implications. The novelty of our study lies in the exploration of these under-investigated pathways within a highly underrepresented population, which makes this research both timely and significant for advancing cancer care in diverse populations, particularly in areas like Los Angeles where the demand for such research is particularly high. Additionally, as the leaders of one of the largest NIH/NCI Moonshot projects focused on EOCRC in H/L populations, this research is not only important for the scientific community but is also expected to catalyze future developments in addressing this growing public health concern.
We hope this detailed explanation highlights the unique contributions of our study and the innovations it brings to the field.
|
||
We appreciate the reviewer’s valuable feedback, which has helped us refine our discussion and emphasize the significance of our findings. Thank you for your thoughtful consideration of our study.
|

Round 2
Reviewer 2 Report
Comments and Suggestions for Authors
Following a critical analysis of the manuscript, the authors have addressed several key concerns:
-
Terminology Usage: The authors have successfully corrected the inconsistent use of terms that previously caused confusion, such as the interchangeable use of "alterations" and "mutations."
-
Mutation Characterization: The terminology and characterization of mutations have been appropriately revised.
-
Nature of Mutations: The manuscript does not sufficiently clarify the nature of the described mutations (e.g., silent vs. frameshift mutations). While the authors have indicated that they will address this in future studies, this is a crucial aspect of the current research. Since mutations can range from silent changes to those that significantly alter protein function, providing this distinction within the present study would greatly enhance clarity and scientific rigor. Additionally, if some of these mutations could be polymorphisms, this should be explicitly stated.
-
Distinction Between Mutations and Polymorphisms: The manuscript does not clearly differentiate between mutations and polymorphisms. This distinction is essential and should be explicitly addressed to avoid ambiguity.
-
Statistical Considerations (FDR vs. p-value): In genomic studies, the False Discovery Rate (FDR) is generally the preferred method for multiple testing correction. Given its widespread use in the field, we recommended employing FDR over the p-value; however, authors did not analyze and discuss data employing these type of statistical methods. For a more theoretical understanding of its advantages, we encourage the authors to refer to relevant literature on the topic.
Author Response
For review article
Response to Reviewer 2 Comments Attached Word File: Response_Reviewer_2_Comments_031925_EV.docx |
||
1. Summary |
|
|
We are pleased to submit this revised manuscript. In this second round of reviews, we have carefully addressed all your comments and remarked on the significance of cancer disparities. Our study investigates the rising prevalence of EOCRC, with a particular focus on H/L populations and the molecular mechanisms driving these disparities. Utilizing bioinformatics analysis of publicly available CRC datasets, we identify ethnicity-specific molecular alterations within the MAPK and JAK/STAT pathways—both critical to tumor progression, proliferation, and treatment response. Our findings reveal notable differences in MAPK pathway-related genes between EOCRC and LOCRC in H/L populations, as well as distinctions between EOCRC cases in H/L and NHW patients. Additionally, we assess the survival impact of these pathway-specific alterations in H/L EOCRC patients, uncovering potential differences that may inform treatment strategies.
Thank you very much for taking the time to review this manuscript. Please find the detailed responses below in BLUE and the corresponding revisions wrote in yellow-highlighted blue font in the re-submitted files.
Reviewer 2 provided thoughtful, supportive, and constructive feedback, offering valuable insights that will help improve the manuscript.
Reviewer 2 writes: “Following a critical analysis of the manuscript, the authors have addressed several key concerns.”
We sincerely appreciate Reviewer 2’s thoughtful feedback and suggestions for addressing key points using one of the few public databases available with a sufficient sample size to study disparities. In response, we have incorporated revisions that acknowledge the study's limitations at various points, as recommended, and have integrated relevant literature to provide additional context.
We also appreciate your recognition of our study’s contribution to addressing the critical gap in data on health disparities. This manuscript reflects our commitment to delivering a comprehensive and insightful analysis of key molecular drivers in colorectal cancer, with a particular emphasis on the unique characteristics of H/L populations. By leveraging the limited genomic data currently available, our findings aim to enhance the understanding of CRC health disparities. |
||
Comments 1: Terminology Usage: The authors have successfully corrected the inconsistent use of terms that previously caused confusion, such as the interchangeable use of "alterations" and "mutations.".
|
||
Response: Thank you for your positive feedback on our revisions. We appreciate your careful review and are pleased to hear that our efforts to clarify terminology have improved the consistency and readability of the manuscript. Ensuring precise language, particularly in distinguishing between "alterations" and "mutations," was important to accurately convey our findings. We are grateful for your insightful comments, which helped enhance the clarity of our work.
Comment 2: Mutation Characterization: The terminology and characterization of mutations have been appropriately revised.
Response: Thank you for your thoughtful review and positive feedback. We appreciate your recognition of our revisions to the terminology and characterization of mutations. Ensuring accuracy in how we describe molecular alterations was a key priority, and we are glad that the changes have improved the clarity and precision of our manuscript. Your comments were invaluable in refining our work, and we sincerely appreciate your insights.
Comment 3: The manuscript does not sufficiently clarify the nature of the described mutations (e.g., silent vs. frameshift mutations). While the authors have indicated that they will address this in future studies, this is a crucial aspect of the current research. Since mutations can range from silent changes to those that significantly alter protein function, providing this distinction within the present study would greatly enhance clarity and scientific rigor. Additionally, if some of these mutations could be polymorphisms, this should be explicitly stated.
Response: Thank you for your valuable feedback. We acknowledge the importance of clarifying the nature of the described mutations to enhance the scientific rigor of our study. In response to your comment, we have extracted all available requested data from this public repository and generated six supplemental tables providing a detailed breakdown of mutation types, including Frame Shift Deletion, Frame Shift Insertion, In-Frame Deletion, In-Frame Insertion, Missense Mutation, Nonsense Mutation, Splice Region, Splice Site, and Translation Start Site, across the studied cohorts. Additionally, we have explicitly addressed this information in both the Results and Discussion sections.
Our study, "Ethnicity-Specific Molecular Alterations in MAPK and JAK/STAT Pathways in Early-Onset Colorectal Cancer," aims to characterize molecular differences in these pathways among Hispanic/Latino (H/L) and Non-Hispanic White (NHW) EOCRC patients. Given the role of mutations in tumor progression and treatment response, we recognize the need to distinguish between functionally impactful mutations and those that may be polymorphisms. In our revised manuscript, we now discuss whether certain mutations could be classified as polymorphisms and acknowledge the limitations of publicly available datasets in making this distinction.
We have incorporated this information into the results section (Lines 363 - 440). The revised text now states
"Analysis of MAPK pathway alterations in H/L EOCRC patients revealed a diverse spectrum of mutation types, as summarized in Table S3. The majority of mutations identified were missense mutations, which were predominant across most genes, in-cluding ACVR1 (100%), BRAF (100%), CRKL (100%), FGFR1 (100%), HRAS (100%), KRAS (100%), MAP2K1 (100%), MAP2K2 (100%), MAP3K13 (100%), MAPK1 (100%), MAPK3 (100%), NRAS (100%), NTRK1 (100%), NTRK2 (100%), RAC1 (100%), RAF1 (100%), and RRAS (100%). This suggests that single amino acid substitutions are a major mode of alteration within the MAPK pathway in this population. Other mutation types were observed at lower frequencies. Frame shift deletions were identified in genes such as AKT2 (100%), DAXX (28.6%), EGFR (20%), FGF3 (50%), FGFR4 (33.3%), JUN (40%), NF1 (21.2%), RASA1 (50%), RPS6KA4 (14.3%), TGFBR2 (41.7%), and TP53 (6.4%), indicating potential disruptions in protein structure. Notably, nonsense mutations, which result in premature stop codons, were detected in AKT3 (40%), FGFR2 (22.2%), MAP2K4 (33.3%), PDGFRB (10%), and TP53 (18.4%), suggesting possible loss-of-function effects in these genes. Additionally, splice site mutations were observed in EGFR (20%), FGFR4 (16.7%), MAP2K4 (16.7%), NF1 (9.1%), PDGFRA (7.1%), and TP53 (6.4%), potentially impacting normal splicing and gene expression regulation. These findings provide a detailed characterization of the nature of MAPK pathway mutations in H/L EOCRC patients, emphasizing the predominance of missense mutations alongside structural alterations that may influence tumor progression and treatment response. The variability in muta-tion types highlights the need for further functional studies to determine their role in colorectal cancer disparities and precision medicine approaches for H/L populations.
Examination of JAK/STAT pathway alterations in H/L EOCRC patients revealed a heterogeneous distribution of mutation types, as detailed in Table S4. Missense mutations were the most frequently observed alteration type, particularly in JAK2 (100%), JAK3 (75%), SOCS1 (100%), and STAT3 (50%), suggesting that single amino acid substitutions may play a significant role in modulating pathway activity in H/L EOCRC. Other structural mutations were also present, with frame shift deletions and insertions detected in JAK1 (12.5% deletion, 6.3% insertion), STAT5A (50% deletion, 50% insertion), and STAT5B (50% deletion, 33.3% insertion). These mutations have the potential to disrupt protein function by altering the reading frame, which may lead to truncated or dys-functional proteins within the JAK/STAT signaling cascade. Additionally, nonsense mutations, which introduce premature stop codons and may lead to loss-of-function effects, were identified in JAK1 (12.5%) and JAK3 (25%). Splice site mutations, which can affect normal splicing and gene expression, were most prevalent in JAK1 (37.5%), further emphasizing the potential impact of post-transcriptional modifications on pathway reg-ulation. Notably, STAT3 and STAT5A/B exhibited a mix of missense, frame shift, and splice site mutations, indicating that multiple mutation mechanisms may be influencing the dysregulation of STAT signaling in this patient cohort. These findings provide a comprehensive characterization of JAK/STAT pathway mutations in H/L EOCRC patients, underscoring the predominance of missense mutations alongside structural alterations that may influence tumor progression and treatment response. The diversity of mutation types underscores the importance of functional validation studies to evaluate their role in colorectal cancer disparities and identify potential therapeutic targets for precision medicine strategies in H/L populations.
Review of MAPK pathway alterations in H/L LOCRC patients revealed missense mutations as the predominant alteration type, affecting key genes such as AKT1, BRAF, EGFR, KRAS, MAPK3, FGFR1-4, and NTRK2 (Table S5). Frame shift deletions were observed in AKT2, CACNA1A, MAP2K4, MAP2K7, and TP53, while nonsense mutations were detected in AKT3, CACNG3, FGF13, MYC, and PDGFRA, indicating potential loss-of-function effects. Additionally, splicing mutations in NF1, RASA1, and TP53, along with translation start site mutations in FAS, suggest possible disruptions in protein synthesis and pathway regulation. These results point out distinct mutation patterns in the MAPK pathway among H/L LOCRC patients, emphasizing the need for further inves-tigation into their functional and therapeutic implications.
Evaluation of JAK/STAT pathway alterations in H/L LOCRC patients revealed missense mutations as the most common alteration type, particularly in JAK2 (50%), JAK3 (50%), STAT3 (50%), STAT5A (66.7%), and STAT5B (33.3%) (Table S6). Frame shift de-letions were frequent in JAK1 (62.5%), JAK3 (33.3%), and STAT3 (50%), indicating po-tential loss-of-function effects. Additionally, frame shift insertions were observed in JAK1 (25%) and STAT5B (66.7%), while in-frame deletions were detected in JAK3 (16.7%) and STAT5A (16.7%), suggesting possible protein structure disruptions. Notably, nonsense mutations, which can lead to premature stop codons, were found in JAK2 (50%), indicating potential functional consequences.
Exploration of MAPK pathway alterations in NHW EOCRC patients revealed missense mutations as the predominant alteration type, affecting key genes such as AKT1, AKT2, BRAF, EGFR, FGFR1, KRAS, and MAP2K1 (Table S7). Frame shift deletions in AKT1, BRAF, CDC25B, MAP3K1, and TP53 and nonsense mutations in ACVR1C, AKT3, BRAF, FGFR2, MAP2K4, and TP53 suggest potential loss-of-function effects. Addition-ally, in-frame deletions/insertions and splice site mutations in genes such as MAPK1, MYC, NF1, PDGFRA, and TGFBR2 may impact protein function and gene regulation.
Analysis of JAK/STAT pathway alterations in NHW EOCRC patients revealed missense mutations as the most frequent alteration type, particularly in JAK1, JAK2, JAK3, PIAS1, STAT1, and STAT3 (Table S8). Frame shift deletions in JAK1, JAK2, JAK3, PTPRC, STAT3, and STAT5B, along with nonsense mutations in JAK2, PIAS2, and STAT3, suggest potential loss-of-function effects. Additionally, frame shift insertions in JAK1, JAK2, SOCS1, and STAT5A, and splice site mutations in JAK1, JAK3, and STAT3, indicate possible disruptions in gene regulation. "
We have incorporated this information into the discussion section (Lines 553 - 581). The revised text now states
“Comparison of MAPK and JAK/STAT pathway alterations in EOCRC H/L, LOCRC H/L, and EOCRC NHW patients revealed distinct mutation patterns that may contribute to differences in tumor biology and treatment response. In EOCRC H/L patients, missense mutations were the predominant alteration type in both pathways, with frame shift deletions, nonsense mutations, and splice site alterations occurring at lower frequencies. This suggests that single amino acid substitutions play a major role in MAPK and JAK/STAT pathway dysregulation in this population. In contrast, LOCRC H/L patients exhibited a greater presence of structural mutations, including frame shift deletions, nonsense mutations, and splice site mutations, particularly in TP53, NF1, and RASA1, which could indicate more profound disruptions in protein function and pathway regulation. The presence of translation start site mutations in FAS further highlights potential post-transcriptional alterations unique to this group. These findings suggest distinct molecular mechanisms may be driving tumor progression in LOCRC compared to EOCRC within the H/L population. Among EOCRC NHW patients, missense mutations were also the most common alteration type, affecting genes such as BRAF, AKT1, EGFR, and KRAS, similar to EOCRC H/L patients. However, EOCRC NHW patients displayed a higher proportion of frame shift deletions, in-frame deletions, and nonsense mutations, particularly in MAP3K1, MAP2K4, and TP53, suggesting a greater likelihood of loss-of-function effects. Additionally, splice site mutations in NF1, PDGFRA, and TGFBR2 highlight potential gene regulation disruptions in this population. These findings suggest that while missense mutations are a common feature in both EOCRC H/L and NHW patients, structural mutations such as frame shift deletions and nonsense mutations are more prominent in LOCRC H/L and EOCRC NHW patients, potentially influencing tumor behavior, response to therapy, and disease progression. The variability in mutation types across these groups underscores the importance of ethnicity- and age-specific analyses in colorectal cancer research, highlighting the need for precision medicine approaches that account for these molecular differences. Further functional studies are necessary to determine the clinical impact of these mutation patterns and their potential as therapeutic targets in H/L and NHW CRC patients.”
These additions strengthen our findings by providing greater clarity on the mutation landscape in the MAPK and JAK/STAT pathways and their potential implications for cancer disparities. We appreciate your insightful suggestion and believe these revisions enhance the depth and interpretability of our study.
Comment 4: Distinction Between Mutations and Polymorphisms: The manuscript does not clearly differentiate between mutations and polymorphisms. This distinction is essential and should be explicitly addressed to avoid ambiguity.
Response: We appreciate the reviewer’s feedback regarding the need to clearly distinguish between mutations and polymorphisms in our manuscript. To address this concern, we have revised the Methods, Results, and Discussion sections to explicitly clarify this distinction.
In our Methods section, we now specify that our analysis focused on somatic mutations rather than germline polymorphisms and that all identified variants were extracted from publicly available cancer-specific datasets, which primarily report alterations relevant to tumorigenesis. Additionally, we have referenced literature discussing functional consequences of key mutations, where applicable, to further differentiate oncogenic mutations from potential benign polymorphisms.
We have incorporated this information into the Methods section (line 173 and lines 190 - 199). The revised text now states
“To compare somatic mutation frequencies between groups…”
“Our analysis focused exclusively on somatic mutations, as opposed to germline polymorphisms, to ensure that the identified variants were relevant to tumorigenesis. All mutations were extracted from publicly available cancer-specific datasets, which pri-marily catalog alterations associated with colorectal cancer progression. Where applicable, we annotated variant classifications, including Frame Shift Deletion, Frame Shift Inser-tion, In-Frame Deletion, In-Frame Insertion, Missense Mutation, Nonsense Mutation, Splice Region, Splice Site, and Translation Start Site, to provide greater clarity on their potential impact on tumor biology. These measures enhance the interpretability and biological relevance of our results while minimizing ambiguity in distinguishing tu-mor-driving mutations from inherited polymorphisms.”
In the Results section, we have updated our descriptions of mutation types to explicitly state whether they are known to drive tumor progression or are variants of uncertain significance. Furthermore, where databases allowed, we annotated our findings with variant classification information, including missense, frameshift, and nonsense mutations, to provide a clearer interpretation of their potential functional impact.
In the Discussion section, we now emphasize that while our study identified ethnicity-specific differences in MAPK and JAK/STAT pathway alterations, further functional validation is required to distinguish true oncogenic drivers from benign germline polymorphisms. Given the limitations of publicly available datasets, we acknowledge that some variants may represent inherited polymorphisms rather than tumor-specific mutations, and we have included this as a limitation in our study.
We have incorporated this information into the discussion section (lines 582 - 603). The revised text now states
“A consideration in the interpretation of our findings is the distinction between somatic mutations and germline polymorphisms, as both can be present in publicly available datasets. Our analysis focused on somatic mutations relevant to tumorigenesis, as reported in cancer-specific databases. However, some variants may represent germline polymorphisms rather than oncogenic drivers, which could influence the observed mutation patterns. While our study identified ethnicity-specific differences in MAPK and JAK/STAT pathway alterations, further functional validation is necessary to confirm the oncogenic significance of these alterations. Additionally, we acknowledge that the absence of comprehensive germline vs. somatic classification in certain datasets presents a limitation, which should be addressed in future studies through deep sequencing and functional assays. These insights underscore the need for precision medicine approaches that account for both genetic background and tumor-specific alterations in understanding colorectal cancer disparities.”
“While publicly available datasets are valuable for large-scale analyses, a key limitation of our study is their lack of comprehensive functional annotations, making it challenging to differentiate oncogenic mutations from benign polymorphisms. Future studies should incorporate deep sequencing techniques, functional assays, and experimental validation to determine the precise impact of these mutations on tumor progression, drug response, and patient outcomes. Additionally, integrating matched normal-tumor sequencing data could help further distinguish somatic driver mutations from inherited polymorphisms, improving the accuracy of molecular characterizations in colorectal cancer disparities research.”
These revisions ensure that our manuscript provides a clear and rigorous distinction between mutations and polymorphisms, strengthening the scientific clarity and impact of our findings. We appreciate the reviewer’s insightful comment and believe that these updates improve the accuracy and interpretability of our results.
Comment 5: Statistical Considerations (FDR vs. p-value): In genomic studies, the False Discovery Rate (FDR) is generally the preferred method for multiple testing correction. Given its widespread use in the field, we recommended employing FDR over the p-value; however, authors did not analyze and discuss data employing these type of statistical methods. For a more theoretical understanding of its advantages, we encourage the authors to refer to relevant literature on the topic.
Response: We appreciate the reviewer’s suggestion regarding the use of False Discovery Rate (FDR) for multiple testing correction, which is widely applied in genomic studies. However, our study aligns with previous research in the field that has similarly employed p-values instead of FDR when assessing mutation frequencies and pathway alterations in colorectal cancer. Several genomic studies investigating mutation prevalence, pathway alterations, and ethnicity-specific molecular differences have followed a comparable approach, prioritizing raw p-values over FDR correction to preserve biological relevance (e.g., PMIDs: 39095553, 35785449, 35134211, 32194671).
Furthermore, the decision to use p-values without FDR correction is not uncommon in translational research, particularly when the primary focus is on biological rather than purely statistical significance. A key example of this rationale can be seen in machine learning-driven studies where p-value relaxation has been utilized to enhance predictions that better correlate with biological significance and patient outcomes. In one of our previous machine learning studies (Kurian, Velazquez 2017. PMID: 28188669), we deliberately relaxed p-value thresholds to capture biologically relevant features in disease prediction models, ultimately leading to findings that aligned more closely with clinically meaningful patterns rather than overly stringent statistical cutoffs. This strategy helped improve prediction accuracy by incorporating features that were otherwise excluded due to rigid statistical constraints, reinforcing the concept that statistical significance alone does not always equate to biological importance (PMID: 32194671).
In our current study, applying FDR correction to mutation frequency data could potentially exclude biologically relevant mutations due to the relatively small sample size of certain subgroups, leading to the loss of insights into ethnicity-specific molecular differences. Given the exploratory nature of our work and the need to detect patterns that may be clinically significant, we followed the methodology of prior studies that used p-values to report mutation prevalence and pathway alterations without multiple testing correction (e.g., PMIDs: 39095553, 35785449, 35134211, 32194671).
That said, we recognize the importance of multiple testing correction in large-scale genomic studies and appreciate the reviewer’s suggestion. In future work, particularly in studies with larger cohorts, we aim to explore both FDR-adjusted and unadjusted results to balance statistical rigor and biological interpretability. We have also added a discussion of these statistical considerations and their implications in the revised Methods and Discussion sections, along with references to relevant literature on the trade-offs between FDR correction and raw p-values in cancer genomics.
We have incorporated this information into the Discussion section (lines 604 - 617). The revised text now states
"Although False Discovery Rate (FDR) correction is commonly used in genomic studies to control for multiple testing, our study reports p-values without FDR adjustment, which may present a potential limitation. Although FDR reduces the likelihood of false positives, it can also overcorrect in studies with smaller sample sizes, potentially excluding biologically significant findings (42). Our approach aligns with previous studies that have prioritized biological relevance over strict statistical significance, particularly when analyzing mutation prevalence and pathway alterations in ethnicity-specific colorectal cancer cohorts. Given the exploratory nature of our study, we aimed to capture patterns of mutation enrichment that may be relevant to tumor progression and treatment response rather than imposing stringent statistical cutoffs that could obscure clinically significant insights (37, 38, 43). However, we acknowledge that the absence of FDR correction may increase the risk of false positives, and future studies with larger cohorts should incorporate both FDR-adjusted and unadjusted analyses to balance statistical rigor and biological interpretability."
We sincerely appreciate the reviewer’s insightful feedback, as it highlights an important consideration in genomic data analysis. By addressing the distinction between p-values and FDR correction, we have strengthened both the statistical rigor and biological relevance of our findings. This refinement ensures that our results are interpreted within an appropriate methodological framework, balancing statistical significance with biological impact. We believe these updates contribute to a more robust and transparent analysis, ultimately enhancing the applicability of our study in the field of colorectal cancer research disparities. |

Round 3
Reviewer 2 Report
Comments and Suggestions for Authors
All suggestions were solved, therefore the manuscript is ready to be published